# Dual-Action Gemcitabine Delivery: Chitosan–Magnetite–Zeolite Capsules for Targeted Cancer Therapy and Antibacterial Defense

**DOI:** 10.3390/gels10100672

**Published:** 2024-10-21

**Authors:** Yuly Andrea Guarín-González, Gerardo Cabello-Guzmán, José Reyes-Gasga, Yanko Moreno-Navarro, Luis Vergara-González, Antonia Martin-Martín, Rodrigo López-Muñoz, Galo Cárdenas-Triviño, Luis F. Barraza

**Affiliations:** 1Laboratorio Térmico de Nano y Macromateriales, Edificio de Procesos Sustentables, Departamento de Ingeniería en Maderas, Universidad del Bío-Bío, Concepción 4081112, Chile; gcardenas@ubiobio.cl; 2Facultad de Ciencias, Departamento de Biología y Química, Universidad del Bío-Bío, Chillán 3780000, Chile; gcabello@ubiobio.cl; 3Instituto de Física, Departamento de Materia Condensada, Universidad Autónoma de México, Coyoacán 04510, Mexico; jreyes@fisica.unam.mx; 4Facultad de Ciencias, Departamento de Química, Universidad de la Serena, La Serena 1720170, Chile; yanko.moreno@userena.cl; 5Departamento de Ciencias Biológicas y Químicas, Facultad de Medicina y Ciencia, Universidad San Sebastián, Campus Las Tres Pascualas, Lientur 1457, Concepción 4060000, Chile; luis.vergara@uss.cl; 6Instituto de Farmacología y Morfofisiología, Facultad de Ciencias Veterinarias, Universidad Austral de Chile, Valdivia 5090000, Chile; antonia.martin@alumnos.uach.cl (A.M.-M.); rodrigo.lopez@uach.cl (R.L.-M.); 7Departamento de Ciencias Biológicas y Químicas, Facultad de Medicina y Ciencia, Universidad San Sebastián, General Lagos 1163, Valdivia 5090000, Chile

**Keywords:** chitosan, magnetic nanoparticles, zeolite clinoptilolite, gemcitabine, drug carriers, ionic gelation

## Abstract

Cancer and infectious diseases are two of the world’s major public health problems. Gemcitabine (GEM) is an effective chemotherapeutic agent against several types of cancer. In this study, we developed macrocapsules incorporating GEM into a chitosan matrix blended with magnetite and zeolite by ionic gelation. Physicochemical characterization was performed using HRTEM-ED, XRD, FESEM–EDS, FT-IR, TGA, encapsulation efficiency (%E.E.), and release profiles at pHs 7.4 and 5.0. Cell viability tests against A549 and H1299 cell lines, and microbiological properties against staphylococcal strains were performed. Our results revealed the successful production of hemispherical capsules with an average diameter of 1.22 mm, a rough surface, and characteristic FT-IR material interaction bands. The macrocapsules showed a high GEM encapsulation efficiency of over 86% and controlled release over 24 h. Cell viability assays revealed that similar cytotoxic effects to free GEM were achieved with a 45-fold lower GEM concentration, suggesting reduced dosing requirements and potentially fewer side effects. Additionally, the macrocapsules demonstrated potent antimicrobial activity, reducing *Staphylococcus epidermidis* growth by over 90%. These results highlight the macrocapsules dual role as a chemotherapeutic and antimicrobial agent, offering a promising strategy for treating lung cancer in patients at risk of infectious diseases or who are immunosuppressed.

## 1. Introduction

Cancer is defined as a broad group of diseases that affect any part of the body and begin with cell death, irregular growth, and subsequent invasion and spread to other organs [1]. The predicted number of cancer cases by 2040 is alarming [2]. Both cellular and tumor-forming cancers have specific treatments, depending on factors such as the cancer stage, the patient’s immune system, interaction with other drugs, tolerance to treatment, and side effects. Among the treatments, chemotherapy is considered an effective but not selective therapy, destroying healthy cells with rapidly dividing characteristics, including hair follicle cells, intestinal epithelial cells, and blood cells [3]. This fact triggers serious effects in patients receiving therapy. In this context, there are chemotherapeutic drugs that act in different ways to produce cell death, inhibiting deoxyribonucleic acid (DNA) synthesis, attacking growth factors, and preventing angiogenesis, among others [4].

Gemcitabine (GEM) is a nucleoside analog chemotherapeutic drug composed of a nitrogenous base and a ribose with two fluorine atoms. It inhibits DNA synthesis by passing through phosphorylation states to the synthesis and blocking the sequence [5]. It is used in monotherapy and in combination with other drugs for the treatment of several major cancers such as pancreatic [6], non-small-cell lung [7], ovarian, breast, colon [8] and bladder [9] cancers. Its biological half-life is estimated to be approximately 17 min, during which it can cause myelosuppression, infection, hemolytic ureic syndrome, skin rashes, alopecia, and respiratory failure, among other effects. However, GEM is active against clinical multi-resistant strains of staphylococcus [10]. In the biological environment it is easily converted into a secondary metabolite, uracil, by deoxycytidine deaminase. The raw material, gemcitabine hydrochloride, is colorless and hydrophilic [11].

New systems have been developed to improve the administration of gemcitabine in the treatment of solid tumors [12], evaluating chemo-resistance and sensitivity factors to improve the efficacy of the therapy [13]. Bijay Singh et al. developed a pharmacological combination of gemcitabine and imiquimod based on hyaluronic acid to stimulate immune cells in the treatment of breast cancer [14]. Shabnam Samimi et al. prepared carbon quantum dots with quinic acid and gemcitabine, tested them in MCF7 breast cancer cells, and obtained interesting results regarding their luminescent properties and high accumulation in tumors thanks to the mixture of these materials [15]. In contrast, Pearl Moharil et al. performed a study on polymeric nanocarriers of gemcitabine decorated with folic acid in combination with doxorubicin for the dual targeting in breast cancer cells and tumor-associated macrophages [16]. On the other hand, Han et al. commented on the different strategies to improve the therapeutic performance of GEM through the fabrication of prodrugs and nanodrugs. The improved systems include liposomes, polymeric particles, inorganic metallic and non-metallic micelles, and nanoparticles, among others [17]. Another example of a novel and possibly effective system is the amphiphilic biodegradable polymeric drug carriers conjugated with gemcitabine developed by Tie-Jun Liang et al., where they obtained promising results from mPEG-PLA/GEM conjugates and showed potential for polymer-based drug targeting and accumulation in tumors, increased the bioavailability and cytotoxic effect on the HT29 cell line [18].

In fact, polymeric matrices with biocompatible and biodegradable characteristics help to improve the bioavailability of these chemotherapeutic agents by avoiding damage to the biological environment (healthy cells) and drug metabolization. One of the most studied polymers due to its interesting properties of solubility in acidic aqueous media, bactericidal properties, biocompatibility, non-toxicity, and compatibility with many other molecules and materials is chitosan (CS) [19,20]. To date, chitosan has been used in several lines of research, including biomedical, agricultural, pharmaceutical, and cosmetic research, among others. Chitosan has also been used to manufacture sustainable packaging (mixed with starch) for the preservation of meats with food safety [21]. Its versatility and modifiability have led to the study of thiolated forms to improve gemcitabine’s loading capability and bioavailability, with results showing enhanced cytotoxic effects compared to pure gemcitabine [22]. It has also been shown that the modified N-trimethyl CS form can significantly improve the oral bioavailability of gemcitabine and its efficacy against breast cancer [23]. Similarly, the inclusion of other chemotherapeutic drugs such as cisplatin or doxorubicin together with gemcitabine in a CS polymeric matrix has also been studied [24,25].

Of course, the conjugation of CS with other materials has not been long in coming to obtain improved gemcitabine loading, delivery, and effectiveness systems. Some of the most common components are metallic and non-metallic particles (silver NPs, iron oxide, silica, zeolites), organic molecules (folic acid, hyaluronic acid), and other biopolymers (PEG, PLGA). These materials readily couple and interact with CS to enhance the bioavailability and targeting characteristics of a delivery system for this drug [26,27].

One of the most studied metallic minerals in combination with CS for drug applications is iron oxide particles [28]. These nanoparticles, in addition to being compatible with chitosan, are approved for use in magnetic resonance imaging (MRI) and have interesting magnetic properties [29]. They increase the potential for targeted and local delivery to target cells and/or tumors in cancer therapy. Likewise, after concentrating in the target tissue, these particles can transform magnetic radiation into local heat and subsequently trigger a process of cell death by a temperature increase (hyperthermia) [30]. Several systems have been studied using functionalized iron oxide nanoparticles as vehicles for gemcitabine loading. The results show the compatibility of gemcitabine with iron oxide nanoparticles and CS, increased uptake, higher accumulation, and the potentiation of gemcitabine cytotoxicity [31,32].

On the other hand, zeolites have unique structural properties, such as a high porosity, an ion exchange capacity, non-toxicity, biocompatibility, and the ability to control their physicochemical properties, which have been studied as scaffolds and potential drug carriers [33,34]. They have antiproliferative and proapoptotic properties on cancer cells [35]. In addition, their affinity with CS results in a synergy of controlled-drug-release properties and subsequent dose adjustment [36,37]. The natural zeolite form, as clinoptilolite. is generally recognized as safe by the Food and Drug Administration (FDA), being on the list of non-carcinogenic minerals, and it becomes a great material to include in the chemo-pharmaceutical delivery system [38].

In order to fill the existing gaps in GEM delivery systems and contribute to improvements in drug bioavailability and delivery, tumor targeting, and anticancer and antibacterial activities, this work focused on developing a novel system. This was carried out using a mixture of materials with specific characteristics, never before put into a composite, in the form of macrocapsules. The importance of including magnetite lies in its magnetic properties and zeolite in its harmlessness, both with possible anchor points and negative charges to generate interactions in the composite. For its part, CS was chosen for its biocompatibility in acidic aqueous solution, which would facilitate potential inner interactions.

The objective of this study was to evaluate the potential of a new delivery system for gemcitabine in the form of CS-based macrocapsules with natural zeolite and magnetite. To ensure the biomedical use of the system and to determine the properties of the components, the initial materials were rigorously characterized. The physicochemical, drug-delivery, microbiological, and cytotoxic properties of the composites were evaluated.

## 2. Results and Discussion

### 2.1. Initial Material Characterization

Initially, the characterization of the initial materials to be included in the composite, namely chitosan, zeolite, and magnetite, that would act as a delivery system for the chemotherapeutic drug was carried out. Table 1 shows the parameters measured in chitosan.

The biopolymer to be used in chemo-pharmaceutical delivery systems must have specific properties to fulfill the therapy-enhancing function. In the case of the chitosan used, with a high molecular weight of 218 kDa, it should avoid the use of any cross-linking agent such as tripolyphosphate (TPP) [39]. The cross-linking of the chitosan chains with this and other agents modifies and slows down the biodegradation of the polymer into smaller molecular units within the biological system. On the other hand, the high degree of deacetylation allows more protonated amino groups in the dilution in the acid medium. By increasing these reactive groups, the interaction with the included materials and the retention of the GEM could be improved.

With respect to the natural zeolite (clinoptilolite), the thermogravimetric analysis (TGA) showed that the percentage mass loss was 11.465% in total and occurred up to approximately 700 °C. As for clinoptilolite zeolite, at higher temperatures (near 1000 °C), thermal stability without mass loss of the material is evident, due to its inorganic nature. The percentage lost is associated with evaporated water molecules, due to their hygroscopic nature. These results of the mass loss of the zeolite agree with those obtained by Korkuna O. et al. [40] and Cerri G. et al. [41], who performed a thermal stability analysis on a sample of natural clinoptilolite. In the micrograph shown in Figure 1A, it can be observed that the zeolite sample presents ordered, agglomerated, crystalline particles (confirmed by X-Ray diffraction—XRD) of various shapes, including flakes, sheets, rods, and prisms [42]. The flake shapes and tubular aggregates coincide with the micrographs reported by the authors in [43]. An average particle size of 1.442 μm was obtained. The energy-dispersive spectroscopy (EDS) analysis in Figure 1B shows the elemental composition and yielded atoms characteristic of the tetrahedral structure of zeolite: Si, Al, and O, and the cation exchange atoms: Na, Ca, Mg, and K. These properties qualify the material for biomedical use [40]. As discussed by the author Servatan M et al. [35], zeolite structures can carry various types of therapeutic agents with high loading efficiency due to their porous structure. In addition to their chemical stability and biocompatibility, they can be modified to achieve stimuli-sensitive carriers for advanced therapies.

The transmission electron microscopy (TEM) micrograph, Figure 1C, shows the microstructure of the material, and in the diffraction pattern Figure 1D, the observed rings match the XRD spectrum shown in Figure 2A, consistent with X-ray tables 00-039-1383 and 00-047-1870 from the PDF database (Powder Diffraction Files). The intensity of the (020), (200), (001), and (021) planes confirms the presence of clinoptilolite-Ca and -Na phases.

The Fourier transform infrared spectroscopy (FT-IR) spectrum in Figure 2B shows bands associated with the stretching vibration of acid hydroxyls at 3626 and 3429 cm^−1^, another band at 1020 cm^−1^ indicating the asymmetric valence of the SiO_4_ tetrahedron, and a confirmation at 471 cm^−1^ with a bending vibration of the Si-O-Si bond.

Magnetite was characterized by its morphology, elemental composition, X-ray diffraction pattern, and characteristic FT-IR bands. Figure 3A shows that the magnetite is in cubes, half-cubes, and half-spheres with average diameter sizes of 154.4 nm, and they are agglomerated. In the EDS spectrum, Figure 3B, carbon appeared due to the composition of the microscope sample holder. Considering the above and normalizing the composition data of magnetite in Fe and O atoms, the elemental composition results in percentages by weight were 62.64% for O and 37.36% for Fe.

The TEM micrograph, Figure 3C,D, shows the microstructure of the material and the diffraction pattern. These data (mostly R5) coincided with the XRD spectrum, Figure 4A, in accordance with X-ray tables 00-019-0629 and 00-039-1346, PDF database (Powder Diffraction Files), where the intensity of 100 of plane (311) confirms that they correspond to the iron oxide phases, magnetite, and maghemite, validating its purity [44]. The FT-IR spectrum, Figure 4B, shows a band at 598 cm^−1^ characteristic of the stretching vibration of the Fe-O bond [45]. Quantification tables of the phases of zeolite and magnetite minerals can be reviewed in Appendix A.

### 2.2. Macrocapsule Characterization

The sample groups outlined below were prepared using the ionic gelation method for subsequent characterization. Several techniques are used to obtain capsules for the biomedical and food industries. In this case, the microencapsulation by ionic gelation proposed by Cárdenas-Triviño et al., 2021 [46], with chitosan, magnetite, and erlotinib, allowed more than 50% encapsulation of the drug, since its solubility depends on pH. In our study, gemcitabine is soluble in water (decreasing the pH), making it compatible with the CS dissolution solution. The formation of the macrocapsules then occurred by the interaction of a polysaccharide and an oppositely charged ion, precipitating the chitosan on the KOH solution. The size of the capsules will basically depend on the diameter of the nozzle or instrument used and the subsequent drying process.

The affinity of the materials for intrinsic characteristics is reflected in the results of interactions to form the composite. Just as chitosan has an affinity for both the individual minerals used and the drug [47], evidence was also found demonstrating that iron ions can be inserted into clinoptilolite frameworks.

#### 2.2.1. Transmission Electron Microscopy and High-Resolution Transmission Electron Microscopy (TEM–HRTEM)

TEM and HRTEM showed that there were no modifications in the crystalline structures of the minerals inside the capsules and in their interaction with the polymeric matrix. To perform this analysis, the particle size of the macrocapsules was reduced by manual crushing with the aid of liquid nitrogen.

Figure 5A corresponds to the CS macrocapsules with zeolite in different concentrations. Figure 5B shows the original forms of the zeolite mineral inside the macrocapsules and Figure 5C shows the structure, atom arrangement, and interplanar distances of zeolite with an average of 0.21 nm (2.1 Å). In addition, Figure 5D shows the CS macrocapsules with magnetite. Figure 5E shows the original agglomerated half-cube shape of the magnetite inside the macrocapsules. Finally, Figure 5F shows the structure with interplanar distances of magnetite (with an average of 0.217 nm).

Regarding the visualization of clinoptilolite by TEM, there have been few studies, since the electron beam that has contact with the sample damages it, and therefore, it is also difficult to obtain the diffraction pattern. However, Yilmaz. S et al. managed to observe natural clinoptilolite loaded with Pd to be used as a catalyst [48]. In the micrograph, overlapping flake-like frameworks can be observed, which are very similar to those obtained in this study. On the other hand, the micrograph obtained by the authors in [49] is a TEM image of agglomerated clinoptilolite with various shapes and sizes. Magnetite, being a crystalline metallic mineral composite material, is more stable, allowing the electron beam emitted by the microscope to pass through it, thus projecting the interference of the light in the micrograph in the form of an image.

For example, Tadic et al. [50] synthesized and characterized iron oxide nanochairs and, with TEM micrographs, were able to see shapes, sizes, and chain lengths, and make segmentations. On the other hand, when performing electrosynthesis and the characterization of iron oxide nanoparticles, the authors in [51] were able to visualize by TEM different shapes, sizes, and agglomerates of these particles. In this study, it was possible to apply HRTEM to see the structure of the minerals and to know the interplanar distances. Comparing the obtained interplanar distance results for magnetite (0.217 nm), they coincide with the data reported by Velsankar V et al., who showed interplanar distances of 0.22 and 0.25 nm in their HRTEM micrographs for nanoparticles obtained from *Echinochloa frumentacea* [52].

#### 2.2.2. Field Emission Scanning Electron Microscopy and Energy-Dispersive Spectroscopy (FESEM–EDS)

Macrocapsules in the form of spheres and hemispheres of about 1.22 mm in diameter were obtained. Rough surfaces and the presence of minerals on the polymeric matrix can be observed in Figure 6. The other micrographs of the remaining macrocapsule groups are shown in Appendix A. In addition, the capability of the instrument reveals, both on the surface and inside the capsule, the particle structures corresponding to the minerals with their characteristic shapes and the shapes of the dry biopolymer resulting from breakage of the chitosan capsule.

EDS analysis revealed compositions corresponding to the structures of CS (C and O), magnetite (Fe and O), and zeolite (O, Si, Al, Ca, K, Na, and Mg). This information is corroborated by the EDS analyses of groups 5, 7, and 14, shown in Appendix A. Potassium was common in most of the analyses, and it is related to the KOH solution in which the capsules were precipitated. The characteristic and distinguishing fluorine atom of GEM was not detected in any system containing the drug.

#### 2.2.3. Fourier Transform Infrared Spectroscopy (FT-IR)

FT-IR analyses were performed in order to prove the loading efficiency of the materials and the GEM in the chitosan macrocapsules. Appendix A shows the bands of groups 3, 5, and 7 with specific components of CS + GEM, CS + Magnetite, and CS + Zeolite, respectively.

Likewise, Figure 7 shows the spectrum of group 14, which in its formulation contains all the materials. Here, the bands at 3368 cm^−1^ and 1030 cm^−1^ correspond to vibrations of the zeolite OH and the asymmetric valence of the tetrahedron. The band at 598 cm^−1^ shows the inclusion of magnetite with the vibration of the Fe-O bond and the band at 1379 cm^−1^ indicates the loading of gemcitabine inside the capsules, with the C-F bond. The other bands correspond to the structure of the chitosan polymeric base.

The broadband peaks between 3000 and 3600 cm^−1^ indicate the incorporation of GEM into the structure of the materials and into possible residues of OH ions, belonging to the KOH precipitation solution [53]. Additionally, in this same section of the spectrum, the vibrations of the -OH bonds in clinoptilolite and the -NH_2_ bonds in chitosan can be observed [54].

#### 2.2.4. Thermogravimetric Analysis (TGA)

A TGA was carried out to evaluate the thermal stability of the macrocapsules. The degradation peaks correspond mainly to CS and GEM (between 315 °C and 284 °C, respectively), since, as observed above, zeolite is stable at higher temperatures. The degradation peak above 400 °C is associated with the total degradation of the organic components and the loss of water from the minerals. The macrocapsule thermogram of group 14 is shown in Appendix A.

For the TGA, the data obtained by the authors in [55] analyzed loaded with an antibiotic (oxolinic acid) formulations. Likewise, they found two maximum degradation temperatures associated with CS (approximately 285 and 590 °C), and the maximum mass loss percentage was 68.09%. In our study, these two temperature peaks were 270.1 and 426.2 °C, and the percentage mass loss ranged from 56.79% to 79.18%.

### 2.3. Gemcitabine Quantification, Encapsulation, and Release Profiles

#### 2.3.1. Gemcitabine Quantification by High-Performance Liquid Chromatography (HPLC)

Initially, a stock standard solution of GEM was prepared at a concentration of 500 μg/mL. Aliquots of 20, 140, 280, 400, 520, 660, and 780 μL were taken from this solution and placed in 25 mL amber volumetric flasks to obtain calibration curve concentrations: 0.4, 2.8, 5.6, 8, 10.4, 13.2, and 15.6 μg GEM/mL, respectively. The samples were then injected into the chromatograph in triplicate, and the GEM peaks (with a retention time of approximately 6.9) were integrated and the area averages were plotted vs. concentrations, obtaining the following equation of the line:(1)$$Y=37.567x-0.646\;R^{2}=0.9998$$

The amount of encapsulated GEM was calculated from the averages of the areas obtained in the chromatograms, in relation to the concentration obtained and the theoretical mass of the drug. Finally, this result of the trapped mass of GEM was applied in the encapsulation efficiency (E.E.%) formula (Equation (2), Materials and Methods). The results in mg GEM and %E.E. were obtained and are arranged in Table 2. These data show that the %E.E. for all formulations is above 82%, evidencing the affinity of both chitosan and minerals with GEM. G10 and G11 formulations displayed similar E.E.% values, though slightly lower than the value shown by G3 (pure CS + GEM), suggesting that the addition of nanomagnetite does not affect the encapsulation capacity of the macrocapsules, giving it useful magnetic properties. In the same way, the G14 formulation, which includes chitosan (CS) and both minerals, exhibited an encapsulation efficiency (E.E.%) of 86.56%. This result suggests that the combination of these minerals with chitosan is compatible and does not hinder its drug-loading capabilities. Additionally, the drug’s aqueous solubility enabled efficient extraction using PBS buffer at pH 7.4, and the HPLC technique effectively separated GEM from the dissolved chitosan due to the pH reduction in the solution.

For the quantification of GEM, the HPLC methodology allowed the correct separation of the peaks and subsequent drug analysis, and the solubility properties of the GEM were visualized from the outset. In the acidic dispersion of the polymer matrix, the amino groups of the GEM are protonated, generating cations such as CS, which can bind to negative groups by electrostatic interaction. The percentages of GEM encapsulated in the systems above 85% show that the technique employed, the form of drug incorporation, and the affinity of the materials favor GEM entrapment. These results are in agreement with those obtained by the authors in [52], who tested the loading of GEM–CS nanoparticles with different amounts of cross-linking agent (Pluronic F127) and obtained GEM loading percentages above 56.22% and up to 71.07%. On the other hand, Kazemi-Andalib et al. fabricated hollow CS and PEG microcapsules for the release of curcumin and GEM, obtaining E.E.% results of 84% and 82% for each drug, respectively [56]. In addition, our results may be influenced by the different concentrations of the minerals.

#### 2.3.2. Release Profiles

Drug release studies were performed to analyze the release kinetics of both free and encapsulated GEM within the macrocapsules. The quantity of GEM released from the dialysis bags was monitored through absorbance measurements at 269 nm using UV–Vis spectroscopy. The data were then normalized to the total amount of the encapsulated drug and presented as a time-dependent percentage of release.

The graph of GEM delivery kinetics in PBS at pH 7.4, Figure 8A, shows an explosion of release of groups 12, 13, and 14, and pure GEM, when at 2 h, they had already released, on average, more than 90%. These results suggest that the encapsulation of GEM in these macrocapsules was rather superficial, since no significant difference was observed with respect to the free drug. However, groups 3, 10, and 11 showed a slower drug release behavior, with the CS + GEM formulation (group 3), releasing GEM the slowest under this pH condition.

To observe the behavior of the samples with some simulated physiological parameters (pH and temperature), delivery profiles were performed. Because the polymeric layer of the macrocapsules (CS) is sensitive to acidic pH levels, it was expected that in PBS solution at pH 7.4, it would take time to be released into the medium [52]. But, this did not occur and means that no potential for pH-directed release was demonstrated in this case. However, for some groups (12, 13, and 14), a burst of GEM release occurred at 2 h (i.e., ~90%). On the other hand, for the groups that contained only magnetite in their formulation, the behavior was more of a controlled release. Similar results were found by Arias. J et al. [57], by testing magnetic CS nanoparticles loaded with GEM with two forms of drug inclusion (absorbed and entrapped). Indeed, under these neutral pH conditions, the trapped GEM had a controlled-release behavior.

For the release in acetate buffer at pH 5.0, only the groups that demonstrated better drug delivery control at pH 7.4 were selected for evaluation. The common factor for groups 10 and 11 is that both contain magnetite (in different concentrations). The kinetics graph at pH 5.0, Figure 8B, shows that the formulations have a controlled drug delivery behavior with respect to pure GEM. These findings can be attributed to the gelation properties of chitosan (CS) under acidic pH conditions, which facilitate a slower release of the drug from the macrocapsules. By 24 h, group 3 (CS + GEM) delivered 58.29%, group 10 (CS + Magnetite 0.0225% + GEM) delivered 87.70%, and finally, group 11 (CS + Magnetite 0.1% + GEM) released 100% in 24 h.

Now, for the experiment with a pH 5.0 solution, the first observation was that after an hour and a half, the macrocapsule content in the dialysis bag began to dissolve. Indeed, the chitosan polymer matrix allowed the release of gemcitabine and in a controlled manner in different delivery percentages for each formulation. From the behavior of groups 10 and 11, it can be inferred that the magnetite concentration influences drug release. Curiously, the formulation that only contained chitosan and gemcitabine was the one that delivered the drug the slowest, approximately 58.29% in 24 h. Importantly, this controlled delivery of the drug at an acidic pH also provides good results for targeting a tumor by other means and then reformulating the doses.

Based on these release results at both pH conditions, these same groups of macrocapsules (3, 10, 11, and 14) were chosen for cell viability assays on lung cancer cells and microbiological assays.

### 2.4. Viability Cellular Assays

The cell viability assay was conducted to compare the cytotoxic effects of GEM-loaded macrocapsules and free GEM on lung cancer cell lines. The extracts from the macrocapsules (groups 3, 10, 11, and 14) contained GEM concentrations approximately 45 times lower (0.022–0.025 mg/mL) than the free GEM solution (1 mg/mL). They were obtained through maceration and trituration with liquid nitrogen. More details can be found in Appendix A. Despite this significantly lower GEM concentration, the results shown in Figure 9A indicate that the macrocapsule extracts exhibited a similar cytotoxic effect to free GEM on A549 cells, with cell viability rates ranging from 74.71% to 84.12% for the macrocapsules, compared to 71.42% for free GEM. This suggests that the encapsulated GEM can achieve comparable anticancer activity with a much lower dosage.

Figure 9B presents the results for the H1299 cell line, where the macrocapsule extracts (specifically G10, G11, and G14) showed even greater cytotoxicity than free GEM, reducing cell viability to 61.59%, 69.97%, and 71.65%, respectively, compared to 75.09% for free GEM. These findings highlight the potential benefits of encapsulation, as it not only maintains but can also enhance the drug’s efficacy. Interestingly, for group 3 (G3), a different trend was observed, with viability decreasing at lower extract concentrations, then increasing slightly at intermediate concentrations (60% and 80%).

Overall, these results demonstrate the efficiency of the encapsulation process, which allows for a sustained release of GEM, resulting in comparable or even superior cytotoxic effects at significantly lower concentrations. This aligns with previous research suggesting that polymeric encapsulation can enhance the cytotoxicity of anticancer drugs through controlled release and increased bioavailability [58]. Notably, groups 10, 11, and 14, which contain magnetite and zeolite materials, exhibited enhanced cytotoxic effects on H1299 cells, indicating that these formulations may provide a more effective treatment strategy for non-small-cell lung cancer (NSCLC).

### 2.5. Microbiological Assays

With the interest of knowing the microbiological properties of the formulations against strains commonly present in hospitals, screening was initially carried out using 10 mg of the ground samples on a solid plate inoculum of *E. coli*, *P. aerugionosa*, *S epidermidis,* and *S. aureus* strains. The MICs and MBCs of all groups of macrocapsules were tested against *S. aureus* and *S. epidermidis*. Groups 10 to 14 showed antibacterial activity (the common components of these groups were magnetite and GEM. As is shown in Table 3, the best activity was achieved against S. epidermidis with MICs ranging between 0.156 and 0.625 mg/mL with similar values for MBCs. The exception was that the MBCs were approximately ten-fold higher (MIC/MBC 0.650/5.0 and 0.156/1.250, respectively). Against *S. aureus*, the MICs were slightly higher, ranging from 0.625 to 1.250 mg/mL, except for group 12 (MIC 0.313 mg/mL). On the contrary, the MBC against *S. aureus* was consistently higher than the MIC. Group 14 was the exception, showing the same value for the MIC and MBC (1.250 mg/mL).

For the *S. aureus* strain, the results are different because, from groups 10 to 13, the MICs are not the same as the MBCs; the two highest concentrations tested (10 and 5 mg/mL) demonstrated a bactericidal effect. However, for group 14, which had all the components in its formulation, the MIC and MBC coincided, and the MBC value was lower than that of the previous groups. This may indicate a synergy of the combination of these materials with inhibitory and bactericidal effects against this strain.

The results obtained in this study showed that the antibacterial activity depends on the components of the formula that contain the drug. Although the synergy of microbiological properties can be inferred, the heterogeneity of the ground powder of the samples must also be considered.. According to Jordheim. L et al. [10], GEM had an inhibitory and bactericidal effect on various strains of multi-resistant staphylococci, varying between 0.08 and 0.25 mg/mL. On the other hand, and no less importantly, the antimicrobial properties of the chitosan itself should be highlighted. For example, a chitosan-based oligosaccharide derivative exhibited an MIC of 2 mg/mL against the *S. Aureus* strain [59]. Of course, the combination of chitosan in this case with the nucleoside analog enhances the bactericidal properties.

In a previous study, we worked on the synthesis and characterization of multifunctional nano-in-microencapsulated systems with the same components and materials, using a spray-drying method, to evaluate the influence of size on the anticancer properties of the composites at smaller scales. The results showed promising systems with high percentages of GEM encapsulation, controlled drug release at an acidic pH, high percentages of magnetic mobility (at 12 mm distance) dependent on the concentration of magnetite, and higher cytotoxicity than pure GEM [1 mg/mL] on A549 and H1299 cells. In lung cancer treatment, aerial administration combined with an external magnetic field to enhance targeting and compound concentration within the tissue could potentially address one of the numerous challenges associated with this chemotherapy [60].

## 3. Conclusions

The initial materials, chitosan, zeolite, and magnetite, have already been used in the biomedical investigation area for various purposes, including as drug delivery systems. While it is true that several systems were found in the literature, this mixture of materials had not been used before, to our knowledge. Their intrinsic characteristics projected a synergy, and for this reason, they were chosen. Also, the use of clinoptilolite-type natural zeolite provides new and possible opportunities for the improvement of systems in drug delivery applications.

On the other hand, the ionic gelation method made it possible to form the capsules and incorporate all the materials. Thanks to the micrographs, it was possible to observe the structures of the minerals inside the capsules and to corroborate their interaction.

The characteristics of the drug (GEM) and its behavior in the acidic dispersion solution of the polymeric matrix (CS) had a significant influence on the high encapsulation percentages. Both these data and those of controlled delivery profiles at pH 5.0 (tumor microenvironment) of the formulations containing CS + nanomagnetite + GEM demonstrated affinity and greater potential to be used as a chemopharmaceutical delivery system.

The results obtained in the cytotoxic efficacy test show a synergistic effect of the encapsulated materials, which improves the cytotoxic response despite the reduced GEM content. Finally, the bactericidal properties of the compounds analyzed improve the antibiotic activity of GEM, presenting lower MIC and MBC values against *S. epidermidis* than in *S. aureus.*

Compared with the microencapsulated systems obtained in the previous study [59], the effect of the size of the capsules that include the same materials can be observed on the encapsulation efficiency and release of GEM, and on the cytotoxic activity against lung cancer cell parameters. The percentages of encapsulation efficiency were between 6 and 10% higher in the microcapsules. Likewise, the delivery profiles showed a lesser degree of controlled release in the macrocapsules at both pHs studied, although macro- and microcapsules show a more sustained release at pH 5 due to the swelling capacity of the CS matrix. Finally, the cytotoxicity tests demonstrated that both the macrocapsules and microcapsules significantly enhanced the cytotoxic effect against the A549 and H1299 cell lines compared to pure GEM. The 100% extract of both systems showed the greatest cytotoxicity, with macrocapsules being particularly effective on H1299 cells, while microcapsules demonstrated strong cytotoxic effects on both A549 and H1299 cell lines.

Macrocapsule composites show significant potential as a delivery system for gemcitabine in the treatment of lung cancer, demonstrating effectiveness against both cancerous cells and bacterial infections. This dual action is particularly important in the context of immunocompromised patients, where the presence of infections can diminish the efficacy of chemotherapy. By effectively targeting cancer cells, such as those in the A549 and H1299 lines, and simultaneously combating bacterial infections, these macrocapsules could enhance the overall therapeutic outcome for patients with compromised immune systems, where infections often exacerbate the progression of cancer and undermine treatment efficacy.

## 4. Materials and Methods

Gemcitabine hydrochloride (purity > 98% by HPLC) was sourced from Sigma Aldrich, (Merck, Santiago, Chile). High-molecular-weight chitosan, with a 96.64% degree of deacetylation and classified as food grade, was acquired from Quitoquímica (Concepción Chile). Clinoptilolite-type zeolite, in 30-mesh powder form, was obtained from Minera San Francisco (San Luis Potosí, México). Iron oxide nanopowder (nano-magnetite) with 97% purity was supplied bySigma Aldrich (Merk, Santiago, Chile). Additional reagents, including KOH, glacial acetic acid, phosphoric acid, sodium acetate trihydrate, and monobasic potassium phosphate were provided by Merck, (Santiago, Chile).

### 4.1. Material Pretreatment and Characterization of Precursor Materials

Zeolite was subjected to particle size reduction treatment using ultraturrax cycles at 10,000 rpm for 8 h, in a volume of 250 mL of water. Subsequently, the solid was recovered by evaporation. Characterization was carried out using high-resolution transmission electron microscopy (HRTEM), electron diffraction (ED), X-ray diffraction (XRD), field emission scanning electron microscopy (FESEM), characteristic X-ray energy division spectroscopy (EDS), Fourier transform infrared spectroscopy (FT-IR), and thermogravimetric analysis (TGA).

CS characterization was performed by molecular mass, degree of deacetylation, TGA, and FT-IR. Meanwhile, magnetite was characterized by HRTEM-ED, XRD, FESEM–EDS, and FT-IR.

### 4.2. Macrocapsule Synthesis by the Ionic Gelation Method

The ionic gelation method is based on generating a gelation reaction between opposite charges, precipitating an acidic solution of CS on an alkaline solution of 0.5 M KOH, drop by drop. The contact of the polycation with the hydroxyl groups (anion) gives rise to the polyelectrolyte reaction, where the outer layer of the CS precipitates. Then, a 4% CS dispersion was prepared in a 1% acetic acid solution and mixed with the amounts described in Table 4 At the same time, a 0.5 mol L^−1^ KOH solution was prepared. The order of addition in the systems containing all the components was GEM, nanomagnetite, and zeolite. The mixtures remained under magnetic mechanical stirring for 8 h at 600 rpm and were then precipitated onto the KOH solution (maintaining the latter’s pH at 13.7 to ensure the formation of the capsules). They were recovered by filtration, washed with distilled water, and checked for a neutral pH. They were dried in a convection oven at 50 ± 2 °C for 5 h [46].

The macrocapsules were designed to contain 2.5% and 0.56% of nanomagnetite and microzeolite, respectively, in relation to chitosan (CS) for both high and low concentrations. For GEM, the intended concentration was set at 0.0625% relative to the CS content.

### 4.3. Macrocapsule Characterization

#### 4.3.1. Transmission Electron Microscopy and High-Resolution Transmission Electron Microscopy (TEM–HRTEM)

A JEOL 2010F microscope with a resolution of 0.19 nm was used. The HRTEM micrographs provided information on the crystalline structure of the materials.

#### 4.3.2. Field Emission Scanning Electron Microscopy and Energy-Dispersive Spectroscopy (FESEM–EDS)

A FEI QUANTA FEG 250FESEM microscope with EDS equipment was used for obtaining the morphologic particle size and chemical composition information of the materials.

#### 4.3.3. Fourier Transform Infrared Spectroscopy (FT-IR)

A Perkin Elmer Spectrum Two infrared spectrophotometer was utilized to examine the incorporation of the various components and the GEM within the macrocapsules. The resulting spectra from the samples were processed and analyzed using the software OriginPro 9.0.

#### 4.3.4. Thermogravimetric Analysis (TGA)

Thermogravimetric analysis was carried out using a TA Instrument model Q50 to assess the thermal stability of the samples. The initial sample weights ranged from 1 to 6 mg, with a constant heating rate of 10 °C/min applied under a nitrogen atmosphere, up to a temperature of 550 °C.

### 4.4. Gemcitabine Quantification, Encapsulation, and Delivery Profiles

For the initial titration from the macrocapsules, the HPLC method (at 275 nm) was used because in the process of maceration, solubilization, and extraction of the drug molecule, the chitosan also showed UV absorbance at the same wavelength. Thus, chromatography was used to separate the peaks and quantify gemcitabine more accurately. On the other hand, the quantification of gemcitabine for delivery profile assays was carried out by UV–Vis spectrophotometry employing a wavelength of 269 nm.

#### 4.4.1. Encapsulation Efficiency (E.E.%)

Gemcitabine quantification was carried out using the HPLC technique (Agilent 1220 Infinity LC), Arquimed, Santiago, Chile. The equipment was set with the following conditions: the mobile phase consisted of a KH₂PO₄ buffer (13.8 g) and H₃PO₄ (2.5 mL) adjusted to pH 2.4 for one liter. The column used was a Zorbax C18; 4.6 × 150 mm; 5-micron. The flow rate was maintained between 1.4 and 1.9 mL/min, and detection was performed at a wavelength of 275 nm. For determining encapsulation efficiency, 20 mg of each sample was measured in triplicate. Each sample was mixed with 1 mL of PBS buffer at pH 7.40, vortexed for 10 min, and the supernatant was filtered through a 0.22 μm filter before injecting 20 μL into the chromatograph. The encapsulation efficiency (E.E.%) was calculated using Equation (2) [60,61]:(2)$$E.\;E.\;\%=\frac{Weigh\;of\;GEM\;entraped\;in\;macrocapsules\;(mg)}{Weigh\;of\;GEM\;add\;initially\;(mg)}\times 100$$

#### 4.4.2. Release Profiles

The drug release studies from the macrocapsules were conducted using the equilibrium dialysis method [62]. For gemcitabine (GEM) release profiles, two different buffer solutions were prepared: PBS at pH 7.4 to mimic physiological conditions and an acetate buffer at pH 5.0 to represent the tumor microenvironment. In summary, 2.0 g of macrocapsules were placed inside a dialysis membrane sac (12.5 kDa cut-off, Merck) containing 3 mL of the prepared buffer. The sealed membranes were then submerged in a beaker with 100 mL of the release medium at 37.0 °C, under continuous agitation at 100 rpm. Aliquots of 700 μL were collected every 5 min for the first 30 min, then at 60 min, and subsequently, every hour up to 6 h. Two additional aliquots were taken at 12 and 24 h. The buffer volume was replenished after each sampling. These samples were analyzed using an Evolution 206 Bio UV–Vis spectrophotometer at a wavelength of 269 nm [63].

### 4.5. Viability Cellular Assays

Cell viability was assessed using the MTT method on immortalized lung cancer cell lines A549 and H1299, seeded at 5000 cells per well in a 96-well plate. Macrocapsule samples from groups 3, 10, 11, and 14 were first macerated with liquid nitrogen. Then, 50 mg of each sample was extracted in RPMI medium (1300 μL) using 0.4 μm inserts in 24-well plates. After a 24 h incubation, the cells were exposed to the sample extracts at concentrations of 100%, 80%, 60%, 40%, and 20%. For the 100% concentration, 100 μL of extract was directly added. For the subsequent dilutions, 800 μL of the extract was mixed with 200 μL of the medium to create the 80% concentration, followed by pipetting 10 times for homogenization. From this mixture, 600 μL was taken and combined with 200 μL of the medium to prepare the 60% concentration. The process was repeated, taking 400 μL and mixing it with 200 μL for the 40% concentration. Finally, 200 μL was mixed with the medium to achieve the 20% concentration, ensuring thorough mixing through pipetting. The estimated GEM concentrations (mg/mL) for each extract are provided in Appendix A.

After 24 h, the MTT reagent was added to stain the viable cells, and the plates were incubated for 4 h at 37 °C. Following this period, the MTT solubilizer was introduced, and the absorbance was measured 24 h later using a TECAN Infinite M Nano UV spectrophotometer at 570 nm.

### 4.6. Microbiological Assays

The antibacterial activity of the different compounds was assayed against *S. aureus* ATCC25923 and *S. epidermidis* ATCC12228. The Minimal Inhibitory Concentration (MIC) was determined by the broth microdilution method according to the EUCAST protocol [64] with minimal variations. Briefly, solutions of 10 mg/mL of each compound were prepared directly in Mueller–Hinton broth. A series of two-fold dilutions between 10 and 0.01 mg/mL were prepared in 96-well, U-shaped-bottom microplates and inoculated with the testing strains at a final concentration of 5 × 10^5^ CFU/mL. To achieve this concentration, an inoculum of 0.5 McFarland (1 to 2 × 10^8^ CFU/mL) was prepared, and from this, appropriated dilutions were performed to obtain de desired concentration. Plates were incubated at 35 °C for 24 h. The MIC was registered as the lowest concentration of the compound that completely inhibited growth. All assays were performed in triplicate.

Minimal Bactericidal Concentrations (MBCs) were determined by inoculating 10 μL from each well without visual turbidity onto a Mueller–Hinton agar plate. The lowest dilution showing no growth was considered as the MBC after 24 h of incubation at 35 °C.

## Figures and Tables

**Figure 1 gels-10-00672-f001:**
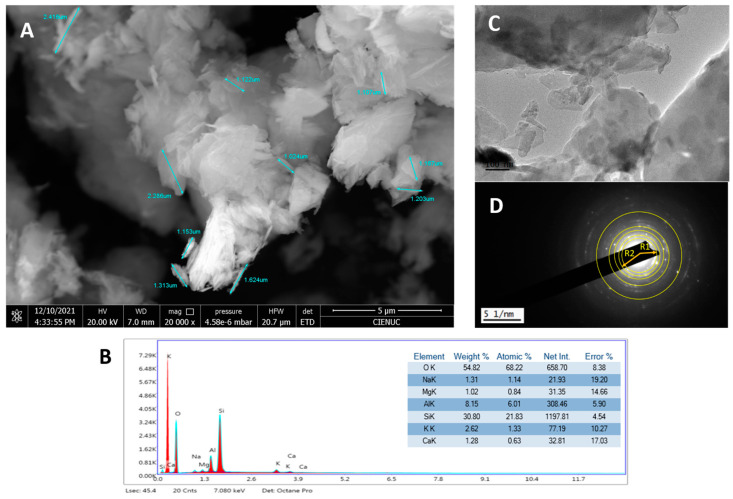
(**A**) FESEM micrograph with 20,000× magnification, (**B**) EDS and elemental composition, (**C**) TEM micrograph, and (**D**) diffraction pattern of zeolite clinoptilolite.

**Figure 2 gels-10-00672-f002:**
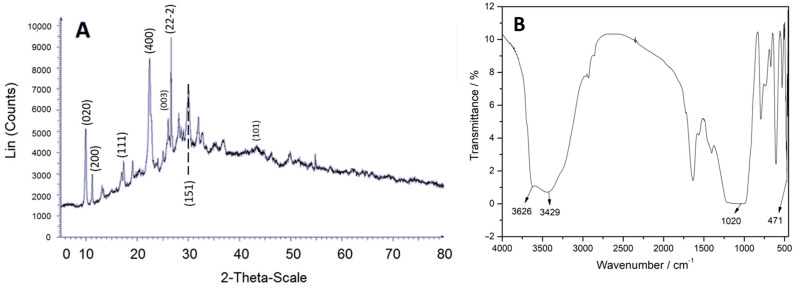
(**A**) XRD planes; (**B**) FT-IR of zeolite clinoptilolite.

**Figure 3 gels-10-00672-f003:**
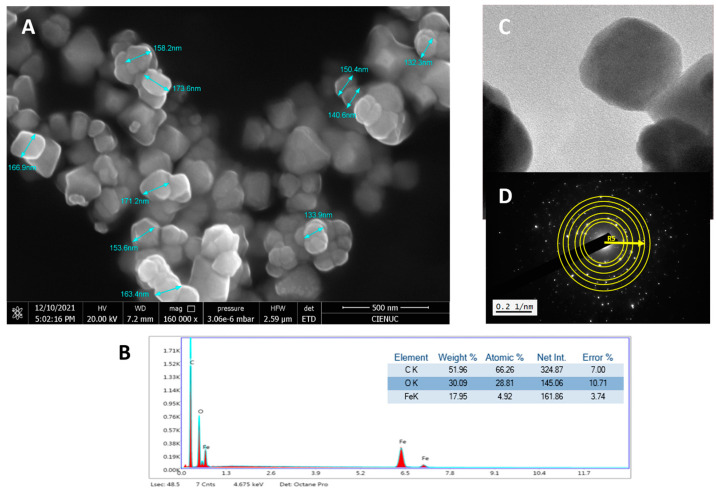
(**A**) FESEM micrograph with 160,000× magnification, (**B**) EDS and elemental composition, (**C**) TEM micrograph, and (**D**) diffraction pattern of magnetite.

**Figure 4 gels-10-00672-f004:**
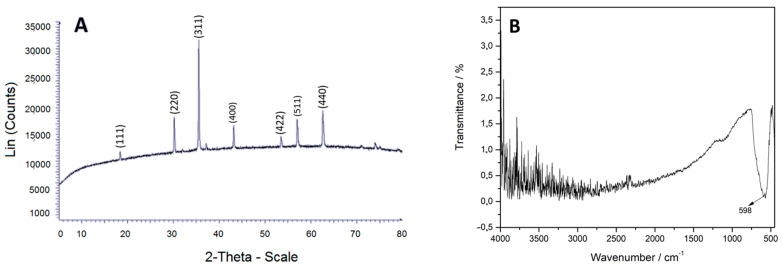
(**A**) XRD planes; (**B**) FT-IR of magnetite.

**Figure 5 gels-10-00672-f005:**
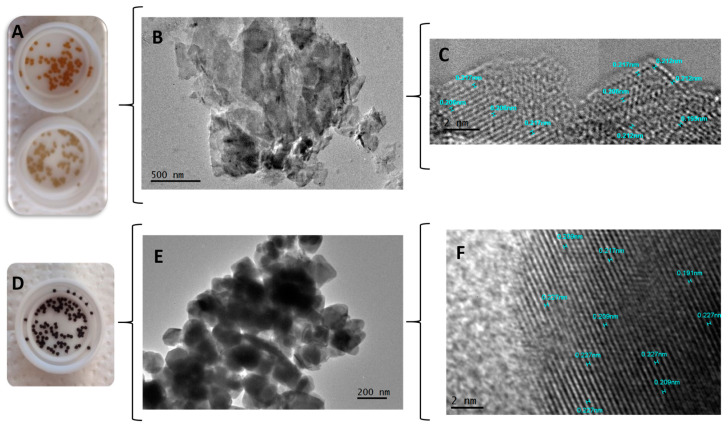
(**A**) Chitosan macrocapsules with clinoptilolite zeolite; (**B**) TEM micrograph of clinoptilolite within chitosan macrocapsules; (**C**) HRTEM micrograph of zeolite; (**D**) chitosan macrocapsules with magnetite; (**E**) TEM micrograph of magnetite within chitosan macrocapsules, and (**F**) HRTEM micrograph of magnetite.

**Figure 6 gels-10-00672-f006:**
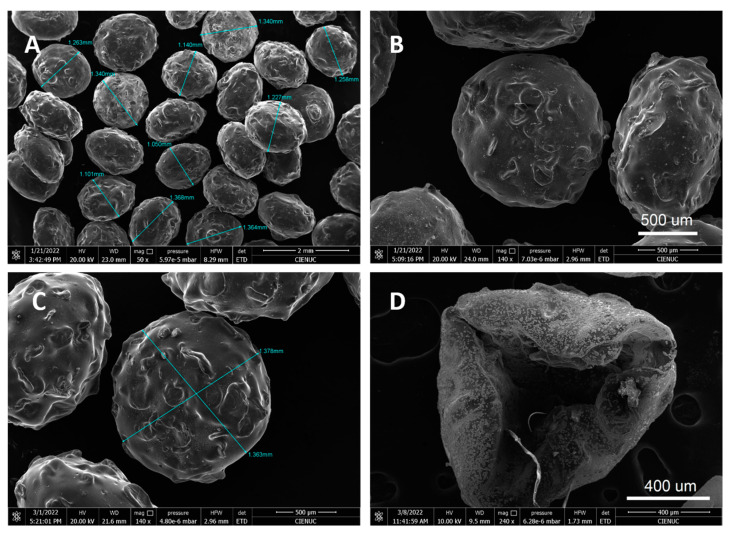
FESEM micrographs; (**A**) macrocapsules, group 2, with 50× magnification; (**B**) microcapsule, group 7, with 140× magnification; (**C**) microcapsule, group 13, with 140× magnification, and (**D**) split microcapsule, group 10, with 240× magnification.

**Figure 7 gels-10-00672-f007:**
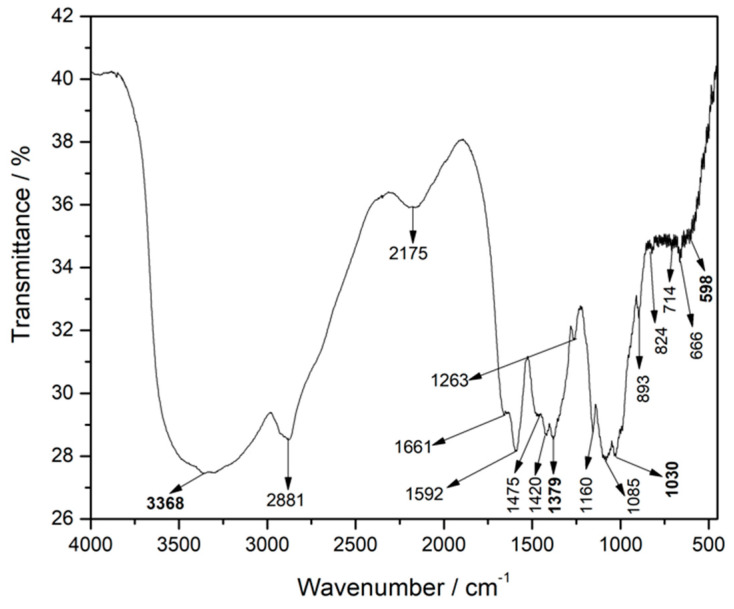
FT-IR spectrum macrocapsules, group 14.

**Figure 8 gels-10-00672-f008:**
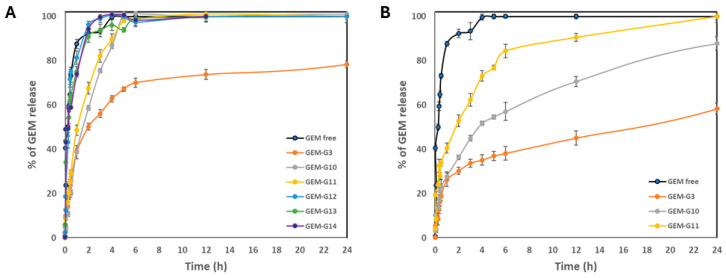
Release profile of free and encapsulated GEM under physiological temperature conditions (37 °C): (**A**) in PBS at pH 7.4 and (**B**) in acetate buffer at pH 5.

**Figure 9 gels-10-00672-f009:**
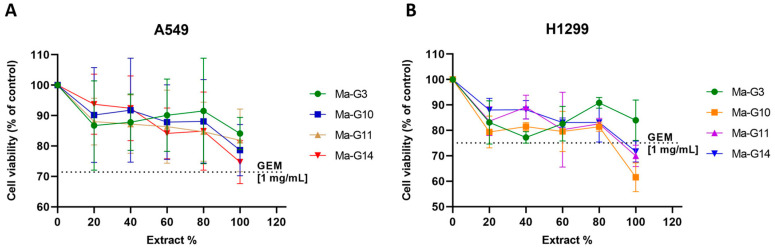
(**A**) Cell viability in A549 and (**B**) in H1299 cell lines.

**Table 1 gels-10-00672-t001:** Chitosan characteristics.

Molecular Mass	Degree of Deacetylation	Thermal Stability	FT-IR	
			λ (cm^−1^)	Functional Group
217,996.17 g/mol	96.64%	Peak of greatest mass lost: 315.47 °C Total mass lost 66.66%.	3430	-OH bond stretching vibration
			2881	-CH bond stretchingvibration
			1647	C=O of Amide I
			1599	NH_2_ group torsion
			1379	Amide III axial deformation
			1160	-O-C-O- group stretching vibration
			604	NH_2_ group deformation

**Table 2 gels-10-00672-t002:** Quantification results of encapsulated GEM (mg) and E.E.% of the macrocapsules.

Group	GEM μg/mL in Supernatant	Total Mass of Initial GEM (mg)	Final Mass of GEM (mg)	Final Mass of Encapsulated GEM (mg)	Efficiency Encapsulation (%)	S.D *
3	0.6473	3.8	0.198	3.602	94.80	0.04
10	1.0765	3.8	0.352	3.448	90.73	0.05
11	1.0063	3.8	0.317	3.483	91.66	0.30
12	1.4347	3.8	0.455	3.345	88.03	1.06
13	1.1081	3.8	0.338	3.462	91.10	0.28
14	1.1188	3.8	0.355	3.445	90.66	0.07

* The standard deviations were calculated from the triplicates of each group.

**Table 3 gels-10-00672-t003:** Summary of MIC and MBC results of gemcitabine-containing macrocapsules.

Samples	STRAINS			
	*S. epidermidis*		*S. aureus*	
	MIC (mg/mL)	MBC (mg/mL)	MIC (mg/mL)	MBC (mg/mL)
Group 10	0.625	0.625	1.250	10.0
Group 11	0.625	0.625	1.250	10.0
Group 12	0.625	5.0	0.313	10.0
Group 13	0.156	1.250	0.625	5.0
Group 14	0.156	0.156	1.250	1.250

**Table 4 gels-10-00672-t004:** Nominal composition of the solution used to prepare macrocapsules.

Group	Macrocapsules	Materials			
		Chitosan	Nanomagnetite	Zeolite	GEM *
2	Chitosan	4.0%	-	-	-
3	Chitosan + gemcitabine	4.0%	-	-	0.0025%
4	Chitosan + nanomagnetite	4.0%	0.0225%	-	-
5	Chitosan + nanomagnetite	4.0%	0.1%	-	-
6	Chitosan + zeolite	4.0%	-	0.0225%	-
7	Chitosan + zeolite	4.0%	-	0.1%	-
8	Chitosan + nanomagnetite + zeolite	4.0%	0.0225%	0.0225%	-
9	Chitosan + nanomagnetite + zeolite	4.0%	0.1%	0.1%	-
10	Chitosan + nanomagnetite + gemcitabine	4.0%	0.0225%	-	0.0025%
11	Chitosan + nanomagnetite + gemcitabine	4.0%	0.1%	-	0.0025%
12	Chitosan + zeolite + gemcitabine	4.0%	-	0.0225%	0.0025%
13	Chitosan + zeolite + gemcitabine	4.0%	-	0.1%	0.0025%
14	Chitosan + nanomagnetite + zeolite + gemcitabine	4.0%	0.0225%	0.0225%	0.0025%

* The amount in mg corresponding to this percentage is 2.55 mg of gemcitabine per 100 mL of solution.

## Data Availability

The data supporting the reported results are contained within the manuscript and the Appendix A.

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
