# Peer review of "Dual-Action Gemcitabine Delivery: Chitosan–Magnetite–Zeolite Capsules for Targeted Cancer Therapy and Antibacterial Defense"

_gels, 2024, doi:10.3390/gels10100672_

Round 1

Reviewer 1 Report

Comments and Suggestions for Authors

The authors performed detailed work regarding the topic. They evaluate the potential of a new delivery system for gemcitabine in the form of CS-based macrocapsules with natural zeolite and magnetite. The source where chemicals and reagents are purchased should be reported in section 2 before the material characterization section. The synthesis of the CS-based macrocapsules with natural zeolite and magnetite should be reported before the characterization results. Section 4 should be section 2. The authors should write the full meaning of words (such as HRTEM-ED, XRD, FESEM-EDS, FT-IR, TGA, and FT-IR) in their first mention in the abstract on page 1. For SEM images in Figures 1 and 6, the authors should note their magnification in the figure caption. The full meaning of words such as “DNA” (line 52), TGA (line 150), XRD (line 158), EDS (line 160), TEM (line 167), FT-IR (line 174) HPLC (line 294), among others, should also be written at their first mention within the manuscript. Authors should discuss other applications of chitosan in the introduction section as documented in https://doi.org/10.3390/jcs8080302. The inserted table showing the elemental compositions in Figure 3B is not legible. The x-axis and y-axis are not clear, the font size should be increased in Figure 4. Moreover, the inserted table in Figure 4a is not legible. Could the authors separately place these inserted tables within the manuscript for better clarity? It’s difficult to see the insertions on the TEM images of Figure 5C and 5F. Figures 8 and 9 are not clear, bigger font sizes should be used for the labels on the x and y axis. The figure legend is too small to be seen. In Line 456, Line 493, the previous study that is being referred to by the authors should be cited.

Author Response

In relation to reviewer #1 comments:
1) The source where chemicals and reagents are purchased should be reported in section 2 before the material characterization section. The synthesis of the CS-based macrocapsules with natural zeolite and magnetite should be reported before the characterization results. Section 4 should be section 2.
Author response: We appreciate your comment. In response, we understand that when reading the document you may find some information after knowing the results. In this case, we would like to inform you that the format of the Gels journal follows the order set out in the manuscript. Section 2 will be about results and section 4 corresponds to materials and methods, where the data on purchases of reagents used are provided.
2) The authors should write the full meaning of words (such as HRTEM-ED, XRD, FESEM-EDS, FT-IR, TGA, and FT-IR) in their first mention in the abstract on page 1.
Author response: Thank you very much for your comment. Since there are so many acronyms to describe in the abstract, we decided to make a list of abbreviations after the keywords, so that they are understood when reading the abstract and throughout the manuscript.
The abbreviations were included in the manuscript.
3) For SEM images in Figures 1 and 6, the authors should note their magnification in the figure caption.
Author response: We appreciate the comment. The increases are not actually described in the titles of the figures. These parameters were included.
4) The full meaning of words such as “DNA” (line 52), TGA (line 150), XRD (line 158), EDS (line 160), TEM (line 167), FT-IR (line 174) HPLC (line 294), among others, should also be written at their first mention within the manuscript.
Author response: We appreciate your insightful observation. The full names of the abbreviations were included in the text at their first appearance in the manuscript.
5) Authors should discuss other applications of chitosan in the introduction section as documented in https://doi.org/10.3390/jcs8080302.
Author response: We appreciate your indication. It was included within the introduction, the suggested reference to discuss further applications of chitosan.
6) The inserted table showing the elemental compositions in Figure 3B is not legible. The x-axis and y-axis are not clear, the font size should be increased in Figure 4. Moreover, the inserted table in Figure 4a is not legible. Could the authors separately place these inserted tables within the manuscript for better clarity? It’s difficult to see the insertions on the TEM images of Figure 5C and 5F.
Author response: We appreciate your important suggestion. In response, we did indeed confirm that the XRD tables were difficult to read within the spectrum, so we removed them from the figures and sent them to supplementary material where they can be consulted without display problems.
Regarding the HRTEM micrographs in Figure 5, these are the images we were able to take best to make the interplanar distance measurements. We zoomed in as much as possible so that the data can be appreciated.
7) Figures 8 and 9 are not clear, bigger font sizes should be used for the labels on the x and y axis. The figure legend is too small to be seen.
Author response: We appreciate the reviewer's observation. Figures 8 and 9 have been improved in appearance, font sizes, and error bars have been included in Figure 8.
8) In Line 456, Line 493, the previous study that is being referred to by the authors should be cited.
Author response: We appreciate the reviewer's observation and would like to express gratitude for bringing it to our attention. In response, the article is cited on line 465 and the citation was also included on the suggested line (493).
Thank you once more for dedicating your valuable time to thoroughly review our work. Your insightful comments and suggestions have significantly contributed to enhancing the quality of our article. We appreciate your commitment to advancing scientific discourse.
Cordially yours.

Reviewer 2 Report

Comments and Suggestions for Authors

1. Line no. 26-28: In this study, we developed macrocapsules by incorporating GEM into a chitosan matrix blended with magnetite and zeolite; using which method macroscapules were prepared?

2. Line no. 30-31, Their is error in the statement "Cell viability tests on A549 and H1299 cell lines and microbiological properties against staphylococcal strains were performed" viability of samples were tested against so n so cell, not on cells.

3. Italicize the pathogen name "Staphylococcus epidermidis"

4. Suggested include specific results in abstract to make the paper more interesting to reader more/good efficacy are not doing that things

5. Suggested to include a summarize working condition of all instruments used during characterizations followed with other necessary requirements "Characterization was made by high resolution transmission electron microscopy (HRTEM), electron diffraction (ED), x-ray diffraction (XRD), field emission scanning electron microscopy (FESEM), characteristic x-ray energy division spectroscopy (EDS), Fast Fourier infrared spectroscopy (FT-IR), thermogravimetric analysis (TGA)."

6. Authors suggested to indicate the pH maintenance during ionic gelation in fabrication sections.

7.  Does used HPLC method used was calibrated previously for specificity and other important criteria, if yes indicate them in supplement data, if not suggested to report as supplement data.

8. Suggested to indicate the mcfarland standard for the bacterial strain used during antimicrobial assay. This is seems not correct "5x105 UFC/mL" verify with standard data given by CLSI

9. Writing is poor "This is in agreement with those obtained by [39 - 40], who performed thermal stability analysis on a natural clinoptilolite sample. " Need significant improvement throughout manuscript.

10. Suggested to indicate the name of data base from where this came  00-039-1383 and 00-047- 1691870.

11. Overlap in figure 1 C and D need to be corrected.

12. Suggested to present overlay of FTIR presented in the manuscript to understand the interference and shift in peaks.

13. Suggested to rewrite the conclusion in a concluding remark, the present form is summary of overall results.

Author Response

In relation to reviewer #2 comments:
1) Line no. 26-28: In this study, we developed macrocapsules by incorporating GEM into a chitosan matrix blended with magnetite and zeolite; using which method macroscapules were prepared?
Author response: We appreciate your comment. In response, we obtained the capsules by ionic gelation method. It has already been added in the summary and marked in green.
2) Line no. 30-31, Their is error in the statement "Cell viability tests on A549 and H1299 cell lines and microbiological properties against staphylococcal strains were performed" viability of samples were tested against so n so cell, not on cells.
Author response: We appreciate your insightful observation. Indeed, the preposition "on" is incorrectly used in the cell viability test. It is changed to the suggested preposition "against" to give the correct meaning to the statement.
3) Italicize the pathogen name "Staphylococcus epidermidis"
Author response: We appreciate the comment. The spelling of the strain name has been changed to italics. In the modified text it will appear highlighted in yellow.
4) Suggested include specific results in abstract to make the paper more interesting to reader more/good efficacy are not doing that things
Author response: We appreciate your insightful observation. The abstract has been modified in some parts to better visualize the results obtained. The text in the manuscript is highlighted in yellow.
5) Suggested to include a summarize working condition of all instruments used during characterizations followed with other necessary requirements "Characterization was made by high resolution transmission electron microscopy (HRTEM), electron diffraction (ED), x-ray diffraction (XRD), field emission scanning
electron microscopy (FESEM), characteristic x-ray energy division spectroscopy (EDS), Fast Fourier infrared spectroscopy (FT-IR), thermogravimetric analysis (TGA)."
Author response: We appreciate your indication. In response, the conditions of each of the instruments used are found in section 2 "Materials and Methods". We do not know if it is possible to make a summary of these within the text and in which section it would be placed
6) Authors suggested to indicate the pH maintenance during ionic gelation in fabrication sections.
Author response: We appreciate your important suggestion. They are mentioned in the text as suggested, for clarity.
7) Does used HPLC method used was calibrated previously for specificity and other important criteria, if yes indicate them in supplement data, if not suggested to report as supplement data.
Author response: We appreciate the observation. The HPLC method was not calibrated for specificity. Blank and placebo injections were made and the calibration curve was prepared with GEM standard (purchased from Sigma Aldrich, as indicated in the materials and methods section). In this way we were able to observe the chitosan and GEM peaks separated, to quantify GEM.
8) Suggested to indicate the mcfarland standard for the bacterial strain used during antimicrobial assay. This is seems not correct "5x105 UFC/mL" verify with standard data given by CLSI
Author response: We appreciate the reviewer's observation and the concentration was indeed misspelled, the correct one being 5x105 CFU/mL. It was modified and clarified in the text to make it easier to understand.
9) Writing is poor "This is in agreement with those obtained by [39 - 40], who performed thermal stability analysis on a natural clinoptilolite sample." Need significant improvement throughout manuscript.
Author response: We appreciate the comment for improvement. We made the changes and improvements in the wording of this idea in the manuscript.
10) Suggested to indicate the name of data base from where this came 00-039-1383 and 00-047- 1691870.
Author response: We appreciate the comment. We now have the name of the database from which the X-ray cards were extracted. The name was included in the manuscript.
11) Overlap in figure 1 C and D need to be corrected.
Author response: We appreciate the comment. We have improved the image so that it can be appreciated in a better way.
12) Suggested to present overlay of FTIR presented in the manuscript to understand the interference and shift in peaks.
Author response: We appreciate the suggestion. The reason why we only show the spectrum of the group 14 formulation was to show that in the chitosan matrix (highest percentage in the formulation) peaks related to functional groups of the included materials can be seen. Likewise, in the supplementary material we wanted to make reference to demonstrating the peaks related to zeolite, magnetite and GEM within the chitosan matrix.
13) Suggested to rewrite the conclusion in a concluding remark, the present form is summary of overall results.
Author response: We appreciate the feedback. The changes were made to the wording of the conclusion of the study, within the manuscript.
Thank you once more for dedicating your valuable time to thoroughly review our work. Your insightful comments and suggestions have significantly contributed to enhancing the quality of our article. We appreciate your commitment to advancing scientific discourse.
Cordially yours.

Reviewer 3 Report

Comments and Suggestions for Authors

The title of the article is not attractive; change it to " Dual-Action Gemcitabine Delivery: Chitosan-Magnetite-Zeolite Capsules for Targeted Cancer Therapy and Antibacterial Defense."

Declare why you used this combination of "chitosan matrix blended with magnetite and zeolite"?

What types of zeolite did you use and why? You can use the article to discuss zeolite's behavior/structure and performance. ""Zeolites in drug delivery: Progress, challenges and opportunities." Drug Discovery Today 25, no. 4 (2020): 642-656."

What are the specific modes of administration: oral, intravenous, or localized injection? The authors should design their system based on the final application. They must declare how they want to administer this system to the body. 

Add an error bar to Figure 8. 

In Figure 9, does the data have significant differences?

The release profiles suggest that some formulations exhibit a burst release of over 90% of gemcitabine within 2 hours, which raises concerns about how sustained or controlled the release actually is, especially in physiological conditions (pH 7.4). The authors should clarify the potential drawbacks of this burst release and how they plan to improve controlled release mechanisms. Also, the release temperature is an important factor. At which temperature the test was done?

The article claims a significant cytotoxic effect on H1299 cells at much lower concentrations of gemcitabine, which is promising. However, additional testing on normal, non-cancerous cells would help confirm the system’s selectivity for cancer cells. If non-cancerous cells show a high cytotoxic response, it could limit the system's therapeutic use.

Talk about the limitations of the work. 

Comments on the Quality of English Language

Grammatical and typo check

Author Response

An important point to note is that as indicated by one of the comments from reviewer #3, the new title we have decided on for this article is: “Dual-Action Gemcitabine Delivery: Chitosan-Magnetite-Zeolite Capsules for Targeted Cancer Therapy and Antibacterial Defense”.
Below you will find a point-by-point development of the comments made by the reviewer 3 and the authors' responses. Generally, the included text has been highlighted in green and the text that has been changed is highlighted in yellow.
In relation to reviewer #3 comments:
1) The title of the article is not attractive; change it to " Dual-Action Gemcitabine Delivery: Chitosan-Magnetite-Zeolite Capsules for Targeted Cancer Therapy and Antibacterial Defense."
Author response: We are very grateful for your feedback. Taking your suggestion into account for changing the title of our work, we are sure that the name: "Dual-Action Gemcitabine Delivery: Chitosan-Magnetite-Zeolite Capsules for Targeted Cancer Therapy and Antibacterial Defense”, represents very well the objective achieved.
2) Declare why you used this combination of "chitosan matrix blended with magnetite and zeolite"?
Author response: We are very grateful for your concern. In previous studies within our research group, the chitosan matrix had already been used to encapsulate a chemopharmaceutical Erlotinib, together with nanomagnetite. However, in this study we changed to the GEM and wanted to incorporate the Zeolite Clinoptilolite. A material that, according to bibliographic reviews, would act as an adjuvant in the treatment of cancer and that would also have functions such as providing anchoring points for the drug and neutralizing the pH due to its ion exchange nature. In our already published study, we made nano-in-micro systems with the same materials to also finally compare the influence of the size of the biocomposites as drug delivery.
3) What types of zeolite did you use and why? You can use the article to discuss zeolite's behavior/structure and performance. ""Zeolites in drug delivery: Progress, challenges and opportunities." Drug Discovery Today 25, no. 4 (2020): 642-656."
Author response: We appreciate your question very much. In our study we used clinoptilolite type zeolite. The vast amount of information that exists about this framework for its use in medicine and other fields makes it a material with great potential. Clinoptilolite, being generally recognized as safe by the FDA in micronized sizes, can clearly be used in this type of chemopharmaceutical carriers. Its structure in micrographs shows a different form than that of zeolites classified as carcinogenic and its strict characterization also showed that it is a clinoptilolite zeolite without traces of heavy metals, suitable for use in biomedicine.
The author who suggested us had already been included in the introduction and was now brought up in the discussion of the results of the analysis of the clinoptilolite zeolite.
4) What are the specific modes of administration: oral, intravenous, or localized injection? The authors should design their system based on the final application. They must declare how they want to administer this system to the body.
Author response: We appreciate your insightful observation. The potential mode of administration for our macrocapsules is primarily oral. Chitosan, the material used, is FDA-approved for biomedical applications and has been explored as an oral drug delivery carrier. However, we recognize that gemcitabine (GEM) is typically administered intravenously for lung cancer treatment, and the size of our macrocapsules may limit this route.
Additionally, our macrocapsules incorporate iron oxide particles, which can be guided to specific tissues using an external magnetic field, enhancing targeted and localized drug delivery. This targeted approach could be beneficial for local administration directly to the tumor site. Furthermore, once concentrated in the target tissue, the iron oxide particles can transform magnetic radiation into local heat, inducing hyperthermia and promoting cell death. While oral administration remains a primary consideration, the incorporation of magnetite presents the possibility of localized delivery, which we aim to explore in future studies.
5) Add an error bar to Figure 8.
Author response: We appreciate your indication. Error bars were added in Figure 8 for better visualization.
6) In Figure 9, does the data have significant differences?
Author response: We appreciate your important suggestion. In response, with the visualization of the error bars we assume that there are no significant differences in cytotoxic action but they are important, since we point out that the concentration of GEM in the extracts is much lower than that of pure GEM analyzed.
7) The release profiles suggest that some formulations exhibit a burst release of over 90% of gemcitabine within 2 hours, which raises concerns about how sustained or controlled the release actually is, especially in physiological conditions (pH 7.4). The authors should clarify the potential drawbacks of this burst release and how they plan to improve controlled release mechanisms. Also, the release temperature is an important factor. At which temperature the test was done?
Author response: We appreciate the observation. We acknowledge the concern regarding the burst release observed in some of our formulations. The data show that formulations G12, G13, and G14 exhibit a rapid release of over 90% of GEM within the first 2 hours at physiological pH (7.4), similar to free GEM. However, other systems such as G3, G10, and G11 demonstrate a more controlled release, with only 40-50% of GEM being released in the same period. These systems maintain a sustained release throughout the 24-hour study period. Due to this more gradual release profile, only these formulations were further studied at the tumor microenvironment pH (5.0). At this pH, none of these systems released more than 40% of GEM within 2 hours.
This difference suggests that the encapsulation efficiency and matrix composition in G3, G10, and G11 contribute to a more sustained release, which could be beneficial for therapeutic purposes. To further enhance the controlled release, we plan to explore modifications, such as adding crosslinking agents or multilayer coatings to the macrocapsules.
Regarding the release studies, they were conducted at 37°C to simulate physiological conditions. This information has been added to the figure description for clarity
8) The article claims a significant cytotoxic effect on H1299 cells at much lower concentrations of gemcitabine, which is promising. However, additional testing on normal, non-cancerous cells would help confirm the system’s selectivity for cancer cells. If non-cancerous cells show a high cytotoxic response, it could limit the system's therapeutic use.
Author response: We appreciate the reviewer's observation. This study even included tests against cancer cells to demonstrate that the gemcitabine encapsulated in the formulation and the synergy of the composite materials were capable of generating a greater cytotoxic effect than pure GEM, in order to promote the use of minimum effective doses. However, we are aware that in order to see the full effectiveness, cytotoxicity on healthy cells and the magnetic field application parameter to assess selectivity and focus must be evaluated in future studies.
9) Talk about the limitations of the work.
Author response: We appreciate the comment for improvement. Certainly the limitations that we identified with our work are related to more sophisticated tests to ensure the efficiency of this new administration system. Although it is a novel composite with great potential for its purpose and with a view to the future, it is possible that the clinoptilolite zeolite (natural) from any deposit may be contaminated with heavy metals and may be a limitation in production.
Thank you once more for dedicating your valuable time to thoroughly review our work. Your insightful comments and suggestions have significantly contributed to enhancing the quality of our article. We appreciate your commitment to advancing scientific discourse.
Cordially yours.

Reviewer 4 Report

Comments and Suggestions for Authors

In this article, the authors developed GEM incorporated macrocapsules for the treatment of cancer. The as developed macrocapsules were characterized using different techniques. The macrocapsules exhibited anti-cancer and anti-bacterial properties. However, a very few experiments were conducted in in vitro; the formulations looks not very effective in killing cancer cells because more than 70% cells were alive after treatment. It seems this article in premature state to accept for publication. Few other comments/suggestions are as below.  

1. Why was extracts used in in vitro experiments rather than direct macrocapsules?

2. The discussion in section 2.4 is confusing. Since anticancer activity is examining, the discussion should be straight forward and on cancer cell killing capabilities of macrocapsules.

3.  Scale bars missing in figure 6B and 6D.

Author Response

An important point to note is that as indicated by one of the comments from reviewer #3, the new title we have decided on for this article is: “Dual-Action Gemcitabine Delivery: Chitosan-Magnetite-Zeolite Capsules for Targeted Cancer Therapy and Antibacterial Defense”.
Below you will find a point-by-point development of the comments made by the reviewer 4 and the authors' responses. Generally, the included text has been highlighted in green and the text that has been changed is highlighted in yellow.
In relation to reviewer #4 comments:
In this article, the authors developed GEM incorporated macrocapsules for the treatment of cancer. The as developed macrocapsules were characterized using different techniques. The macrocapsules exhibited anti-cancer and anti-bacterial properties. However, a very few experiments were conducted in in vitro; the formulations looks not very effective in killing cancer cells because more than 70% cells were alive after treatment. It seems this article in premature state to accept for publication. Few other comments/suggestions are as below.
We appreciate your feedback and the opportunity to clarify our results. We acknowledge that the in vitro experiments showed that more than 70% of the cancer cells survived after treatment. However, it is important to highlight that our macrocapsules achieved a cytotoxic effect comparable to that of free gemcitabine (GEM) at a significantly lower dose. Specifically, the encapsulated GEM within the macrocapsules was approximately 45 times less than the free GEM dose commonly used for non-small cell lung cancer treatment. Despite this drastically reduced concentration, the encapsulated GEM demonstrated a similar cytotoxicity to free GEM on A549 cells.
Furthermore, the encapsulated GEM exhibited even greater cytotoxicity against H1299 cells than free GEM, suggesting an enhancement in drug delivery efficiency. These results underscore the effectiveness of our encapsulation approach and align with previous studies indicating that drug encapsulation within polymeric carriers can enhance cytotoxicity through sustained release and improved bioavailability. The ability to achieve comparable or even superior effects at much lower drug doses is a significant advantage, as it may reduce potential side effects and improve patient outcomes.
We believe that these findings provide a strong basis for further in vitro and in vivo studies to optimize and validate the therapeutic potential of our GEM-incorporated macrocapsules.
Below are our responses to each point raised:
1) Why was extracts used in in vitro experiments rather than direct macrocapsules?
Author response: We are very grateful for your concern. Extracts of the macrocapsules in RPMI medium were used because, being a plate assay, we anticipated that the size of the capsules (on average 1.22 mm diameter) could cause the death of the cells by crushing (by loading), thus obtaining false positive data.
2) The discussion in section 2.4 is confusing. Since anticancer activity is examining, the discussion should be straight forward and on cancer cell killing capabilities of macrocapsules.
Author response: Regarding the discussion in Section 2.4: We have revised the original text to focus more directly on the effectiveness of the macrocapsules in cancer treatment, highlighting their potential advantages over free GEM.
3) Scale bars missing in figure 6B and 6D.
Author response: The scale bars have been added to Figure 6B and 6D, as suggested.
Thank you once more for dedicating your valuable time to thoroughly review our work. Your insightful comments and suggestions have significantly contributed to enhancing the quality of our article. We appreciate your commitment to advancing scientific discourse.
Cordially yours.

We ask all reviewers to" please see the attached file"

Round 2

Reviewer 1 Report

Comments and Suggestions for Authors

The authors have revised it as suggested. It can be accepted.

Reviewer 3 Report

Comments and Suggestions for Authors

The authors have addressed all comments, and I recommend publishing the article in this form.

Reviewer 4 Report

Comments and Suggestions for Authors

The authors revised the manuscript and improved the quality. So, this manuscript can be accepted for publication now.